# Multiscale structural and electronic control of molybdenum disulfide foam for highly efficient hydrogen production

Jiao Deng[1,2], Haobo Li[2], Suheng Wang[1], Ding Ding[1], Mingshu Chen[1], Chuan Liu[1], Zhongqun Tian[1], K.S. Novoselov[3], Chao Ma[4], Dehui Deng[1,2] & Xinhe Bao[2]

Hydrogen production through water splitting has been considered as a green, pure and high-efficient technique. As an important half-reaction involved, hydrogen evolution reaction is a complex electrochemical process involving liquid-solid-gas three-phase interface behaviour. Therefore, new concepts and strategies of material design are needed to smooth each pivotal step. Here we report a multiscale structural and electronic control of molybdenum disulfide foam to synergistically promote the hydrogen evolution process. The optimized three-dimensional molybdenum disulfide foam with uniform mesopores, vertically aligned two-dimensional layers and cobalt atoms doping demonstrated a high hydrogen evolution activity and stability. In addition, density functional theory calculations indicate that molybdenum disulfide with moderate cobalt doping content possesses the optimal activity. This study demonstrates the validity of multiscale control in molybdenum disulfide via overall consideration of the mass transport, and the accessibility, quantity and capability of active sites towards electrocatalytic hydrogen evolution, which may also be extended to other energy-related processes.

[1] State Key Laboratory of Physical Chemistry of Solid Surfaces, Collaborative Innovation Center of Chemistry for Energy Materials (iChEM), College of Chemistry and Chemical Engineering, Xiamen University, Xiamen 361005, China. [2] State Key Laboratory of Catalysis, Collaborative Innovation Center of Chemistry for Energy Materials (iChEM), Dalian Institute of Chemical Physics, Chinese Academy of Science, Dalian 116023, China. [3] School of Physics and Astronomy, University of Manchester, Oxford Road, M13 9PL Manchester, UK. [4] Center for High Resolution Electron Microscopy, College of Materials Science and Engineering, Hunan University, Changsha 410082, China. Correspondence and requests for materials should be addressed to D.D. (email: dhdeng@dicp.ac.cn) or to X.B. (email: xhbao@dicp.ac.cn).

The properties of two-dimensional (2D) $MoS_2$ are significantly different in comparison with its three-dimensional (3D) form. Thus, it has been considered for a number of applications, such as solar cells[1–3], photocatalysis[4–6], lithium ion batteries[7–9] and electrocatalysis[10–13]. Owing to its natural abundance, low cost and good catalytic performance, recently $MoS_2$ has become a representative non-precious material for electrocatalytic hydrogen evolution reaction (HER) of water splitting[14–21]. Such liquid-to-gas electrochemical conversion, with a complex reaction process at the interface of liquid ($H^+$), solid (catalyst) and gas ($H_2$), require a multiscale structural and electronic control of $MoS_2$ to make each involved reaction step to proceed smoothly. This includes sufficient transport of reactants and products, accessibility of catalyst surface, abundant active sites and enough catalytic capability. Similar to the recent developments in the mesoporous framework of graphene[22–24] or polymer[25,26] foam, the design and preparation of a uniform mesoporous $MoS_2$ foam could simultaneously facilitate the mass transport and accessibility of active sites. Yet, unlike the flexibility of carbon atoms skeleton in graphene and organic small molecules in polymers, $MoS_2$ with single-crystal layer composed of three molecular layers (S–Mo–S) appears much more inflexible, which leads to such engineering still remaining a great challenge. Also, the S-edges of 2D $MoS_2$ is usually considered as the active sites, while the in-plane structure is not active in catalysis[15,27–31]. Our recent work demonstrated that introducing different dopant atoms into the $MoS_2$ matrix can enhance the intrinsic activity of its in-plane S atoms[32]. Therefore, further atomic-scale engineering via doping hetero atoms into the mesoporous $MoS_2$ foam may achieve a multiscale modulation to synergistically boost the HER electrochemical process. However, such all-round structural and electronic control within $MoS_2$ to enhance the HER performance has not been reported before.

Herein, we present a multiscale structural and electronic control of $MoS_2$ foam for highly efficient HER process: (i) the macro-scale: a uniform mesoporous $MoS_2$ foam (mPF-$MoS_2$, average pore size $\sim 30$ nm) facilitate the transport of $H_3O^+$ and $H_2$, and increases the accessibility of $MoS_2$ surface; (ii) the nano-scale: oriented vertical growth of $MoS_2$ nanosheets around the mesopores increase the number of edges as the active sites; (iii) the atomic-scale: further chemical doping with transition metal Co atoms into the mPF-$MoS_2$ framework enhance the intrinsic HER activity (mPF-Co-$MoS_2$). Such mPF-Co-$MoS_2$ electrocatalyst exhibits an excellent durability and a low overpotential of only 156 mV at the current density of 10 mA cm$^{-2}$, comparable to the most active $MoS_2$-based HER electrocatalysts in acidic medium (Supplementary Table 1). Furthermore, the density functional theory (DFT) calculations confirmed the experimental results that an appropriate Co doping content can greatly promote the HER activity of $MoS_2$. The strategies, introduced in the present work, may open new opportunities for the rational design of $MoS_2$ through a multiscale structural and electronic control to strengthen the electrocatalytic HER and other energy-related process, and possibly for the structural control of other 2D materials.

## Results

**Synthesis of mesoporous $MoS_2$ foam.** The uniform mesoporous $MoS_2$ foam (mPF-$MoS_2$) was prepared with the synthetic procedure illustrated in Fig. 1. First, $(NH_4)_6Mo_7O_{24}$ molecules were homogeneously adsorbed onto the colloidal $SiO_2$ nanospheres via a wet impregnation method. Then, the direct chemical reaction with $CS_2$ on $SiO_2$ surface was conducted to convert Mo precursors into small $MoS_2$ domains. Because of the induction of the monodisperse $SiO_2$ nanospheres template, these small

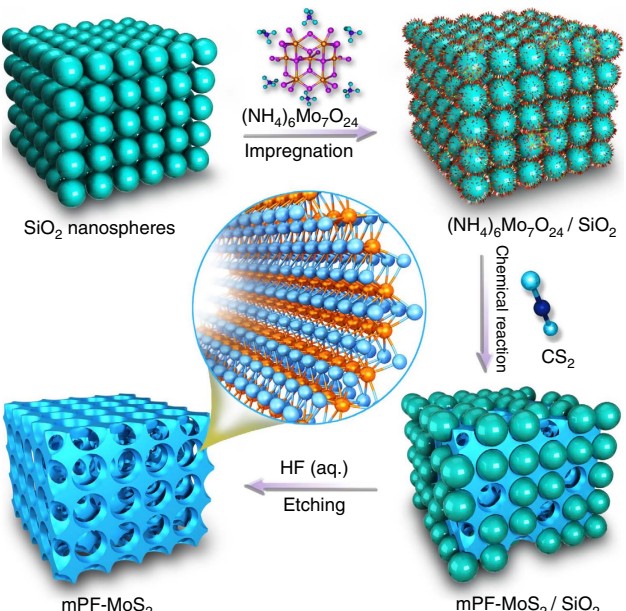

**Figure 1 | Schematic illustration of the fabrication of mesoporous $MoS_2$ foam.** The direct chemical synthesis was adopted with the $(NH_4)_6Mo_7O_{24}$ and $CS_2$ as precursors, assisted by the colloidal $SiO_2$ nanospheres.

domains would further self-assembly into vertically aligned $MoS_2$ layers around the $SiO_2$ nanospheres. Finally, the mPF-$MoS_2$ can be obtained via etching the $SiO_2$ template with HF solution. Note that the etching process will not influence the structure of $MoS_2$ because the $MoS_2$ can not be dissolved by HF solution (Supplementary Fig. 1).

**Structural analysis of mesoporous $MoS_2$ foam.** The scanning electron microscopy (SEM) and transmission electron microscopy (TEM) as well as high-angle annular dark field-scanning transmission electron microscopy (HAADF-STEM) show that the mPF-$MoS_2$ possessed abundant spherical voids derived from the residual spaces after the removal of $SiO_2$ nanospheres, leading to a uniform porous framework (Fig. 2a–d, Supplementary Figs 2 and 3a). Note that these uniform nanopores are interconnected throughout the entire 3D $MoS_2$ foam at different orientations by 3D tomography (Supplementary Figs 4 and 5, Supplementary Movies 1 and 2), which facilitates the mass transport and accessibility of active sites during the catalytic process. The energy-dispersive X-ray (EDX) maps exhibits that the Mo and S elements were distributed homogeneously in the porous framework (Fig. 2e). The $N_2$ adsorption-desorption isotherms indicate the presence of mesopores with a narrow pore size distribution at $\sim 24$ nm (Fig. 2f).

The high resolution (HR) TEM image shows a typical interlayer distance of 0.62 nm corresponding to the (002) plane of $MoS_2$ (Fig. 2g), and the hexagonal 2H-$MoS_2$ crystal characteristics could also be gain from the X-ray diffraction (XRD) pattern (Fig. 2h). Remarkably, the $MoS_2$ layers were almost vertically aligned around the mesopores with a large fraction of exposed edge sites (Fig. 2g and Supplementary Fig. 3b). Compared with random-oriented $MoS_2$ nanosheet (rNS-$MoS_2$) sample prepared without $SiO_2$ template, the mPF-$MoS_2$ showed no obvious difference in the XRD patterns (Fig. 2h), Raman spectra (Supplementary Fig. 6), X-ray photoelectron spectroscopy (XPS) (Supplementary Fig. 7) and X-ray absorption near-edge structure (XANES) spectra (inset of Fig. 2i). But according to the HRTEM images comparison

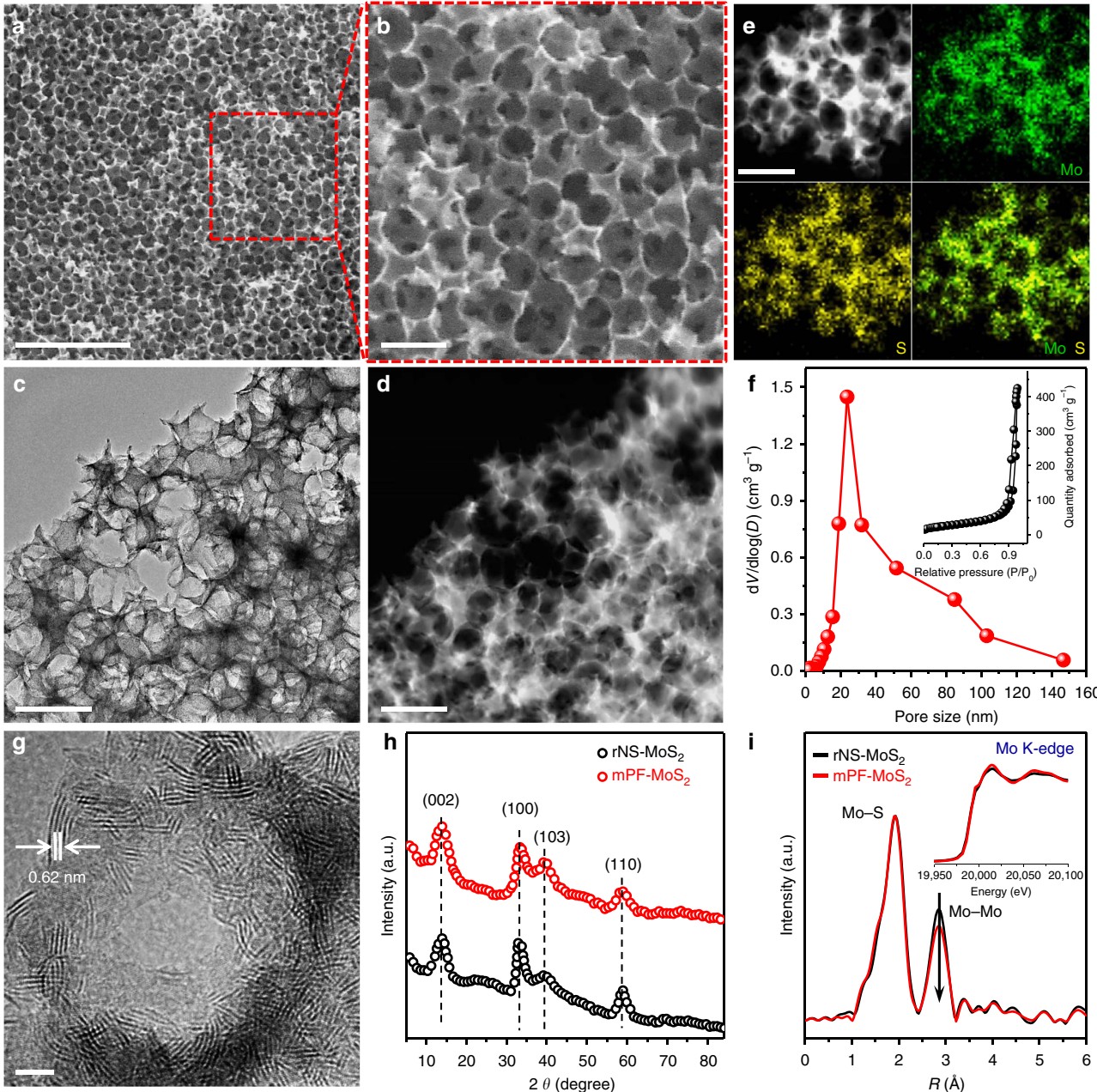

**Figure 2 | Morphology and structural characterizations of mesoporous MoS₂ foam.** (**a,b**) SEM images of mPF-MoS₂. (**c,d**) TEM image and corresponding HAADF-STEM image of mPF-MoS₂ at the same position. (**e**) HAADF-STEM image and corresponding EDX maps of mPF-MoS₂. (**f**) Pore size distribution and N₂ adsorption–desorption type IV isotherms (inset) of mPF-MoS₂. (**g**) HRTEM image of mPF-MoS₂ with inset showing a typical MoS₂ layer distance of 0.62 nm and a distinct mesopore. (**h**) XRD pattern of mPF-MoS₂ in comparison to rNS-MoS₂. (**i**) The $k^2$-weighted EXAFS spectra of mPF-MoS₂ in comparison with rNS-MoS₂. The inset is the normalized Mo K-edge XANES spectra of mPF-MoS₂ in comparison to rNS-MoS₂. Scale bar: (**a**) 500 nm, (**b**–**e**) 100 nm, (**g**) 5 nm.

(Supplementary Fig. 3), the mPF-MoS₂ with vertical aligned layer and smaller lateral size possessed much more exposed edge sites. In addition, the extended X-ray absorption fine structure (EXAFS) spectra (Fig. 2i) exhibited that mPF-MoS₂ had less Mo-Mo coordination than rNS-MoS₂, also confirming the mPF-MoS₂ possessed more edge sites. This should increase the catalytic activity of mPF-MoS₂ significantly.

**Electrocatalytic performance of mesoporous MoS₂ foam.** A typical three-electrode setup in 0.5 M H₂SO₄ electrolyte was adopted to conduct the electrocatalytic measurements. Bulk MoS₂

shows a poor HER activity with only a minor improvement observed for rNS-MoS₂ (Fig. 3a). From Fig. 3b, one could see that compared with rNS-MoS₂, the required overpotential to drive a HER current density of 10, 20 and 50 mA cm⁻² within mPF-MoS₂ reduced 195, 219 and 262 mV, respectively. Particularly, the overpotential at a current density of 10 mA cm⁻² for mPF-MoS₂ is 210 mV, which is superior to the reported dense vertically aligned MoS₂ film[30]. Furthermore, the mPF-MoS₂ showed a long-term stable performance within the accelerated degradation measurements by 5,000 cyclic voltammetric (CV) sweeps (Fig. 3c), which also indicated mPF-MoS₂ is a good non-precious alternative for HER electrocatalyst.

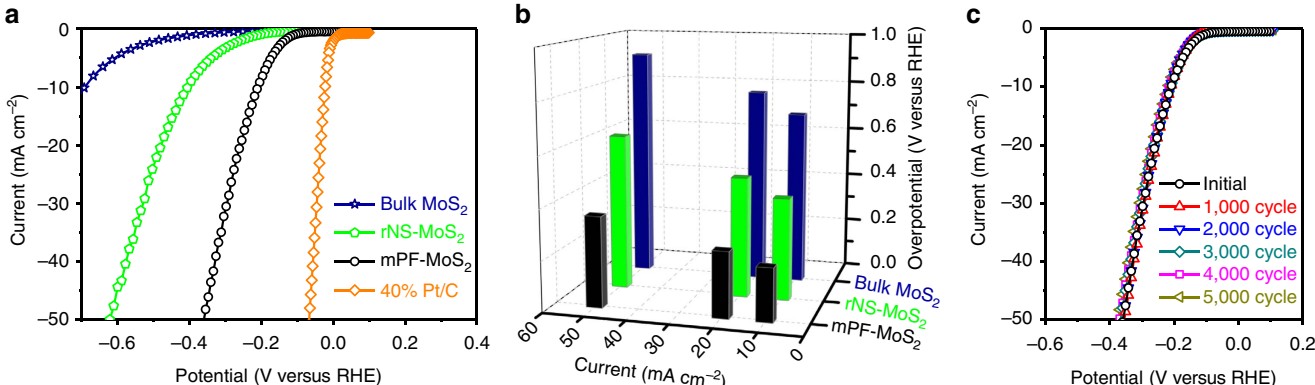

**Figure 3 | Electrocatalytic HER performance of mesoporous MoS$_2$ foam.** (**a**) HER polarization curves for mPF-MoS$_2$ in comparison with bulk MoS$_2$, rNS-MoS$_2$ and 40% Pt/C. (**b**) Overpotential at current density of 10, 20 and 50 mA cm$^{-2}$ for mPF-MoS$_2$ compared with rNS-MoS$_2$ and bulk MoS$_2$. (**c**) Durability measurement of mPF-MoS$_2$. The polarization curves were recorded initially and after every 1,000 sweeps between $-0.1$ and $+0.5$ V (versus RHE) at 100 mV s$^{-1}$. All the HER measurements were conducted in an Ar-saturated 0.5 M H$_2$SO$_4$ electrolyte at 25 °C.

It is usually considered that only the edge sites of pure MoS$_2$ own the HER activity, while the basal plane is catalytically inert[14,15]. Therefore, the mPF-MoS$_2$, possessing rich vertical edge sites, is expected to be more catalytically active. Moreover, numerous mesopores in mPF-MoS$_2$ facilitate the mass transport. Meanwhile, according to the contact angle measurements (Supplementary Fig. 8), the mPF-MoS$_2$ (23°) become more hydrophilic relative to the rNS-MoS$_2$ (32°) and bulk MoS$_2$ (105°), leading to the more easy accessibility of reactants on active sites for the mPF-MoS$_2$ catalyst. In addition, the massive mesopores with curved surface in MoS$_2$ 2D plane may induce the strain, which can further increase the electrocatalytic activity referring to the literatures[19,33].

**Chemical doping of mesoporous MoS$_2$ foam.** Doping of different transition metal atoms into the MoS$_2$ matrix can enhance the intrinsic activity of its in-plane S atoms[32]. Here, we introduced Co atoms into the mPF-MoS$_2$ framework by *in situ* adding Co precursors within the impregnation procedure (see the experimental section for details), yielding a Co-doped mesoporous MoS$_2$ foam (mPF-Co-MoS$_2$). As shown in the SEM (Supplementary Fig. 9), TEM (Supplementary Fig. 10a) and HAADF-STEM (Fig. 4a) images, the mesoporous MoS$_2$ foam has been well retained after Co doping, with MoS$_2$ flakes still assembling as vertically aligned layers around the mesopores (Supplementary Fig. 10b). The Co dopants bring indiscernible chemical state variation of the MoS$_2$ framework according to the XPS spectra (Supplementary Fig. 11) and Mo K-edge XANES spectra (Supplementary Fig. 12). No Co-containing nanoparticles were observed from TEM images, consistent with the EDX maps showing the homogeneous distribution of Co, Mo and S elements over the entire mesoporous framework (Fig. 4a).

Co doping contents within the mesoporous MoS$_2$ foam can be easily modulated by varying the amount of Co precursors, resulting in a series of mPF-Co-MoS$_2$-x samples (x represents the Co doping contents in wt.%). As shown in the Co K-edge XANES spectra (Fig. 4b) and EXAFS spectra (Fig. 4c), all Co atoms in different mesoporous MoS$_2$ foam possess the valence and the Co–S bonds are distinguished from those in commercial CoS crystal. This indicates that Co atoms are covalently doped into the MoS$_2$ 2D plane rather than being adsorbed on the surface. This finding is also confirmed by the Mo K-edge EXAFS spectra (Fig. 4d) showing a decrease of Mo-Mo coordination caused by the substituted-doping of Co atoms within the MoS$_2$ 2D plane. In

addition, the decrease of Mo-Mo coordination accompanied with the increase of Co doping contents (Fig. 4d) was also consistent with the stepwise red shift of E$^1_{2g}$ and A$_{1g}$ modes in Raman spectra (Fig. 4e) resulted from the progressively increased Co dopants in MoS$_2$ 2D plane to soften the Mo-S related modes and decrease their vibration frequency[34]. Nevertheless, the E$^1_{2g}$ and A$_{1g}$ modes of MoS$_2$ will change significantly when the Co doping contents exceeded 16.7% (Fig. 4e), suggesting a structural variation in the mesoporous MoS$_2$ foam. This correlates with the XRD patterns showing that the crystal structure of MoS$_2$ was well maintained with no other phases appearing after Co doping, until the Co doping contents were 21.1% or more (Fig. 4f). Meanwhile, distinct change in pore structure of mPF-Co-MoS$_2$ samples appeared when the Co doping contents exceeds 16.7% (Supplementary Fig. 13). Note that mesoporous MoS$_2$ foam with different Co doping contents showed no obvious difference in the contact angle measurements (Supplementary Fig. 14). The above analyses indicated that there is an optimum doping content (16.7% from our experience) which will provide significant Co contents but still preserving the mPF-Co-MoS$_2$ integrated mesoporous vertically aligned framework.

**Effect of Co dopant on electrocatalytic performance.** In view of the additional atomic-scale modulation in mesoporous MoS$_2$ foam, a further enhanced HER process was expected. Thus, mPF-Co-MoS$_2$-3.4 exhibited a distinctly enhanced activity, reducing the overpotential of 26 and 53 mV at the current density of 10 and 50 mA cm$^{-2}$ relative to mPF-MoS$_2$ (Fig. 5a). The sample with Co doping content of 16.7% demonstrates the optimum activity (Fig. 5b). The volcano-shaped relationship between HER activity and Co doping contents confirmed our finding that there is an optimum Co doping level which, from one hand enhances the intrinsic catalytic activity of mPF-MoS$_2$ and at the same time maintains the inherent framework within mesoporous MoS$_2$ foam. Remarkably, the mPF-Co-MoS$_2$-16.7 showed a high HER activity with the overpotential at a current density of 10 mA cm$^{-2}$ of only 156 mV (Fig. 5a), comparable to the most active MoS$_2$-based non-precious HER electrocatalysts in acidic medium (Supplementary Table 1). Moreover, mPF-Co-MoS$_2$-16.7 also showed a long-term stable performance even after 5,000 CV sweeps within the accelerated degradation measurements (Fig. 5c). Tafel plots showed that mPF-Co-MoS$_2$-16.7 with a Tafel slope value of 74 mV dec$^{-1}$ and mPF-MoS$_2$ (90 mV dec$^{-1}$) followed a similar reaction process via

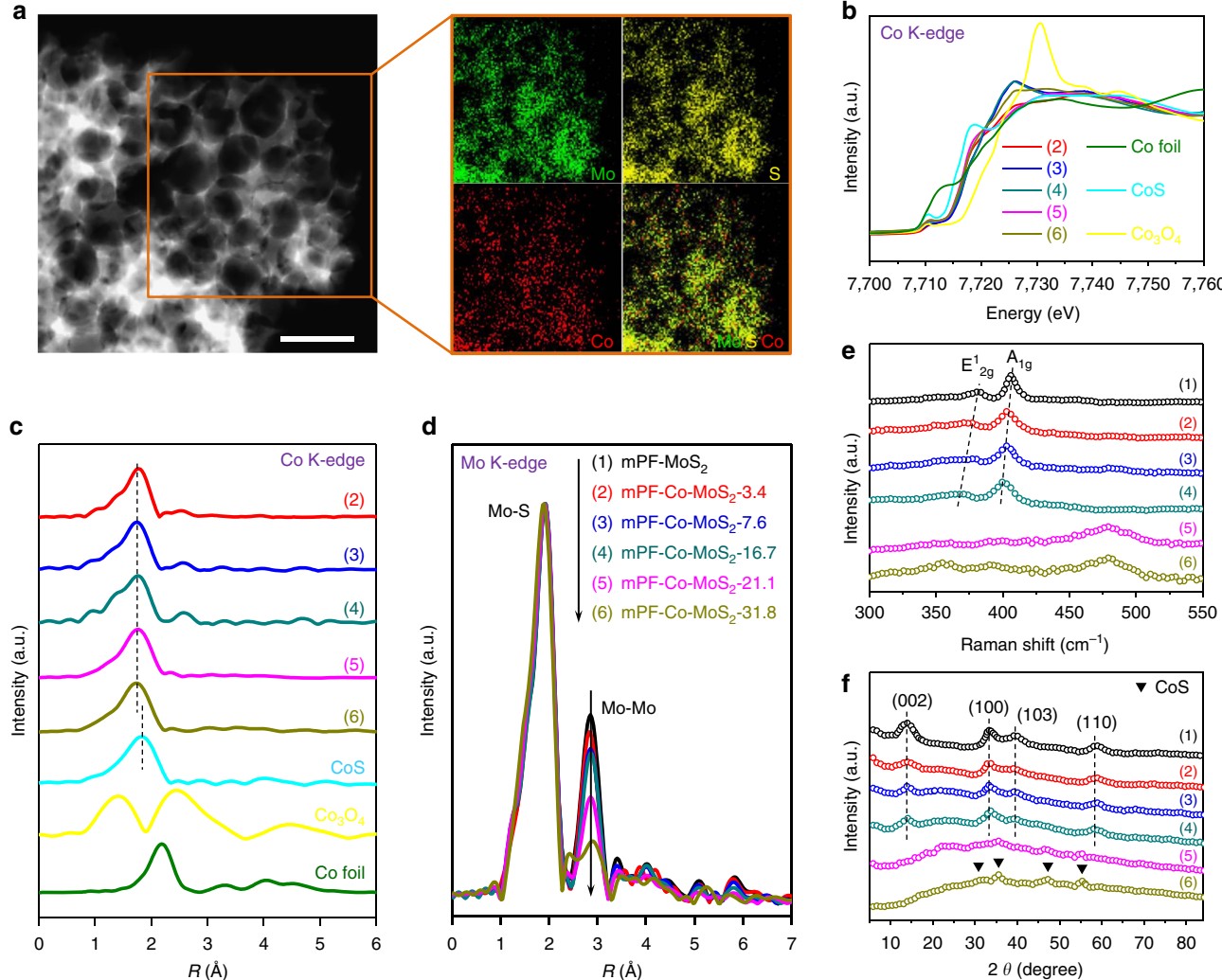

**Figure 4 | Structural and electronic properties of various Co-doped mesoporous MoS₂ foam.** (**a**) HAADF-STEM image and corresponding EDX maps with orange rectangle in HAADF-STEM image of mPF-Co-MoS$_2$-16.7. Scale bar, 100 nm. (**b**) Co K-edge XANES spectra of a series of mPF-Co-MoS$_2$ samples in comparison to Co foil, CoS, and Co$_3$O$_4$, respectively. (**c**) Co K-edge $k^2$-weighted EXAFS spectra of a series of mPF-Co-MoS$_2$ samples in comparison with CoS, Co$_3$O$_4$ and Co foil, respectively. (**d**) Mo K-edge $k^2$-weighted EXAFS spectra of various mPF-Co-MoS$_2$ samples compared with mPF-MoS$_2$. (**e**) Raman spectra of different mPF-Co-MoS$_2$ samples in comparison to mPF-MoS$_2$. (**f**) XRD patterns of a series of mPF-Co-MoS$_2$ samples in comparison with mPF-MoS$_2$. The numbers (1), (2), (3), (4), (5) and (6) represent mPF-MoS$_2$ and mPF-Co-MoS$_2$ with the Co doping contents of 3.4, 7.6, 16.7, 21.1 and 31.8%, respectively.

the Volmer–Heyrovsky mechanism[35–37], deviating from the Pt/C electrocatalyst (30 mV dec$^{-1}$) via the Volmer-Tafel mechanism (Fig. 5d). These results demonstrated that the Co doping content will significantly affect the activity modulation of MoS$_2$, and a moderate value can maximally promote the multiscale structural and electronic control in mesoporous MoS$_2$ foam for the HER activity optimization.

**Theoretical studies of Co doping effect**. DFT calculations were carried out to gain further insights into the influence of different Co doping contents within the basal plane of MoS$_2$ on the HER activity. The hydrogen adsorption free energy ($\Delta G_H$) is a widely accepted descriptor of HER activity for various catalytic materials, where the optimal value of $\Delta G_H$ is around zero ($\sim$0) eV to compromise the reaction barriers and achieve the best HER activity[38,39]. For the basal plane of pristine MoS$_2$, the $\Delta G_H$ is $\sim$2 eV, far away from the optimal value. With Co atoms introduced into the MoS$_2$ in-plane, taking the coverage ($\theta_H$) of

1/4 monolayer (ML) as an example, the $\Delta G_H$ decreased continuously and reached $\sim$0 eV at the Co doping content of 13.3 wt.% (atomic ratio of Co:Mo is 1:2), beyond which the $\Delta G_H$ will depart away from the optimal values again (Fig. 6a). These simulations indicate that there indeed exists a moderate Co doping content to promote MoS$_2$ to gain the optimal HER activity, confirming the experimental results.

To understand the origin of the increased HER activity with increased Co doping content, an analysis of the electronic properties has been made. First, the projected density of state of S atoms show a significant increase in the electronic states of in-plane S sites around Fermi level after Co atoms doping (Supplementary Fig. 16), resulting in the enhanced catalytic activity, in accordance with our previous study[32]. Furthermore, according to the molecular orbital theory, when H atom is absorbed on surface S atom, the combination of H 1s orbital and S 3p orbital will form a bonding orbital ($\sigma$) and anti-bonding orbital ($\sigma^*$), where the degree of energy level matching between H atom and S atom determines the H–S bonding strength (Fig. 6b).

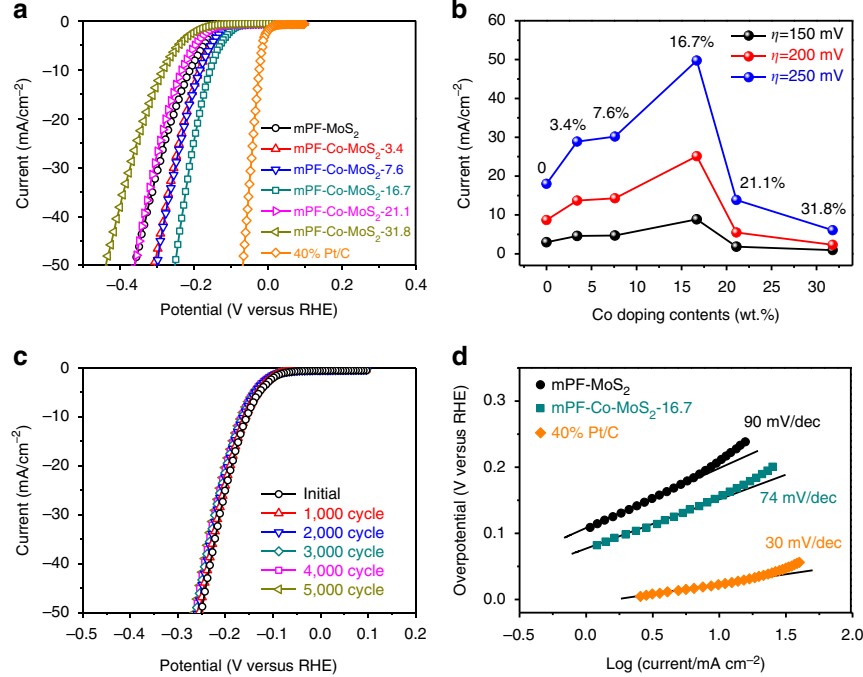

**Figure 5 | Effect of Co doping on the HER performance of mesoporous MoS₂ foam.** (**a**) HER polarization curves for mPF-Co-MoS₂ with different Co doping contents in comparison with mPF-MoS₂ and 40% Pt/C. (**b**) Current densities at overpotential of 150, 200 and 250 mV for mPF-Co-MoS₂ with different Co doping contents compared with mPF-MoS₂. (**c**) Durability measurement of mPF-Co-MoS₂-16.7. The polarization curves were recorded initially and after every 1,000 sweeps between − 0.1 and + 0.5 V (versus RHE) at 100 mV s⁻¹. (**d**) Tafel plots for mPF-MoS₂, mPF-Co-MoS₂-16.7 and 40% Pt/C, respectively. All the HER measurements were conducted in an Ar-saturated 0.5 M H₂SO₄ electrolyte at 25 °C.

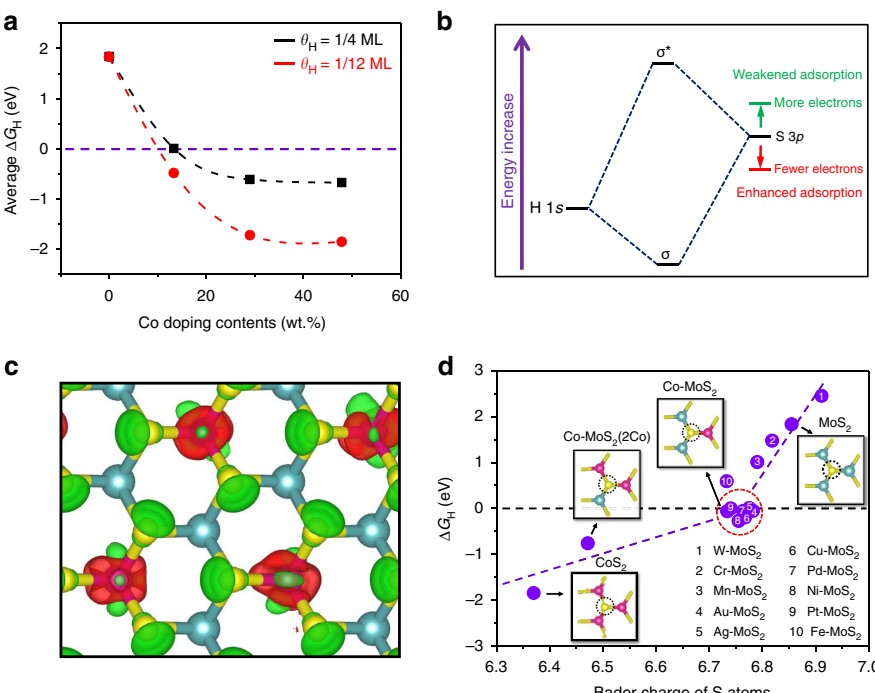

**Figure 6 | Theoretical calculations for the effect of Co doping contents on HER of MoS₂.** (**a**) Average $\Delta G_H$ on S atoms versus the Co doping contents, considering different coverage of 1/4 ML and 1/12 ML. The corresponding optimized catalyst structures can be seen in Supplementary Fig. 15. (**b**) Schematic diagram of the bonding of H 1s orbital and S 3p orbital (from MoS₂), where depletion of electrons on S atoms will lower the orbital position and enhance the H–S bond. (**c**) Differential charge density of Co-doped MoS₂ (Co doping content of 13.3 wt.%, Co:Mo atomic ratio of 1:2). Red and green contours represent electron accumulation and depletion, respectively. The isosurface level is set to be 0.11 e/Bohr³. (**d**) $\Delta G_H$ on S atoms versus the Bader charge of S atoms for different structures, with the detailed data for each point shown in Supplementary Table 2. The insets are the atomic configurations of one S atom bonding with three Co, two Co and one Mo, one Co and two Mo, as well as three Mo atoms, respectively. Green balls: Mo; yellow balls: S; pink balls: Co.

Because of the very high energy level of S $3p$ orbital relative to H $1s$ orbital, the H adsorption on basal plane of pristine $MoS_2$ is too weak ($\Delta G_H = \sim 2\,eV$), leading to a poor HER performance of $MoS_2$ for HER. When doping another metal atom such as Co into the $MoS_2$ in-plane, the electron number on S atom will decrease (Fig. 6c) to offset the energy level mismatching for enhancing the H adsorption and HER activity. Different metal atoms own the different capability to modify the electron density on S atoms (see Fig. 6d and Supplementary Table 2 for detailed data), and the metal dopants that tune the Bader charge of S atoms into the range of $\sim 6.73$ to $\sim 6.78$ will lead to a moderate $\Delta G_H$ (Fig. 6d) and high HER activity (Supplementary Fig. 17). Among them, the Co atom is indeed a good regulator to bring the $\Delta G_H$ get $\sim 0\,eV$. Nevertheless, further increasing Co doping contents will cause the excessive decrease of electron on S atoms to make the interaction between H atoms and S atoms too strong (Fig. 6d). In addition, high Co doping contents can also lead to less stable of $MoS_2$ surface according to the surface energy ($\gamma$) calculations (Supplementary Fig. 18), which is also harmful to the HER activity.

## Discussion

In summary, we introduce a multiscale structural and electronic control of $MoS_2$ strategy to achieve the high-efficient HER electrocatalysis. First, a uniform mesoporous $MoS_2$ foam (mPF-$MoS_2$) was fabricated with a significantly enhanced HER performance compared with that of random-oriented $MoS_2$ nanosheet (rNS-$MoS_2$). It originates from the macro-scale modulation fabricating massive mesopores to gain the sufficient transport of $H_3O^+$ and $H_2$, the favourable accessibility of $MoS_2$ surface and the strain-induced promotion, as well as the nano-scale modulation with vertically aligned layers to provide abundant active edge sites. Second, chemical doping was introduced to add further atomic-scale engineering in the mesoporous $MoS_2$ foam for the intrinsic activity increase. The optimum Co-doped mesoporous $MoS_2$ foam (mPF-Co-$MoS_2$) with a Co content of 16.7% showed a further distinct enhancement of HER activity, which possessed a long-term durability with more than 5,000 recycles and an overpotential of only 156 mV at the current density of 10 mA cm$^{-2}$, comparable to the most active $MoS_2$-based electrocatalysts in acidic medium. DFT calculations confirmed the experimental results that a moderate Co doping content can modulate the H adsorption on $MoS_2$ to a suitable degree and simultaneously maintain the structure stability to promote the HER activity reach optimum value. The findings in the present work pave a rational pathway to strengthen the electrocatalytic HER performance of $MoS_2$ via the multiscale structural and electronic control, and the involved concept and strategy can be extended to other energy-related process or other 2D materials.

## Methods

**Materials synthesis.** The mPF-$MoS_2$ was synthesized through a direct chemical synthesis method. First, 400 mg $(NH_4)_6Mo_7O_{24} \cdot 4H_2O$ and 5,333.4 mg $SiO_2$ colloidal disperse (30 wt.% $SiO_2$ in ethylene glycol, Alfa Aesar) were dispersed in 20 ml deionized water, followed by stirring under room temperature to remove the solvent and drying under 80 °C. Then, the gained solid and 10 ml $CS_2$ were transferred into a 40 ml stainless steel autoclave under Ar and maintained at 400 °C for 4 h. The final product was treated with HF (aq.) under room temperature for 5 h, followed by washing with water and absolute ethanol for several times and drying at 80 °C. For comparison, the rNS-$MoS_2$ was synthesized by using 900 mg $(NH_4)_6Mo_7O_{24} \cdot 4H_2O$ dissolved in 20 ml deionized water and 10 ml $CS_2$ conducted within the same chemical reaction as the mPF-$MoS_2$ without using $SiO_2$ template. The series of mPF-Co-$MoS_2$ samples were synthesized by using 400 mg $(NH_4)_6Mo_7O_{24} \cdot 4H_2O$, specified amount of $Co(NO_3)_2 \cdot 6H_2O$ and 5,333.4 mg $SiO_2$ colloidal dispers to gain the impregnated solid, and then with 10 ml $CS_2$ to proceed within the same process as the mPF-$MoS_2$. The Co doping contents in final

mPF-Co-$MoS_2$ samples were measured by inductively coupled plasma atomic emission spectroscopy.

**Materials characterization.** SEM was conducted on Hitachi S4800 operated at 20 kV. TEM, HAADF-STEM and EDX mapping were carried out on a FEI Tecnai 30 microscope and a 20 microscope operated at an accelerating voltage of 300 and 200 kV, respectively. The 3D tomography in the STEM mode was carried out on the FEI Talos F200 × microscope operated at 200 kV. $N_2$ adsorption–desorption was measured with a Micromeritics Tristar 3020 Surface Area and Porosimetry analyzer. XRD measurements were conducted on a Rigaku Ultima IV diffractometer with Cu K$\alpha$ radiation at 35 kV and 15 mA. XANES and EXAFS were measured at the BL14W1 beamline of the Shanghai Synchrotron Radiation Facility (SSRF). Raman spectroscopy was performed on a Renishaw inVia Raman microscope with a 532 nm excitation laser at a power of 0.29 mW. XPS measurements were carried out on an Omicron XPS System used Al K$\alpha$ X-rays as the excitation source with a voltage of 15 kV and power of 300 W. Contact angle of water solution droplet on the surface of catalyst layer were conducted on SDC-100 contact angle measurement instrument (Shengding Precision Instrument Co., Ltd., China) at room temperature. Inductively coupled plasma atomic emission spectroscopy was carried out in Varian AA240z graphite furnace atomic absorption spectrometer.

**Electrochemical measurements.** HER polarization curve tests were conducted on a Princeton Parstat MC potentiostat/galvanostat with a three-electrode electrochemical cell equipped with a gas flow controlling system. Graphite rod was used as the counter electrode and Ag/AgCl (saturated KCl-filled) as the reference electrode. A glassy carbon rotating disk electrode with a diameter of 5 mm covered by a thin catalyst film was used as the working electrode. Typically, 4 mg catalyst was suspended in 1 ml ethanol with 20 μl Nafion solution (5 wt.%, Du Pont) to form a homogeneous ink assisted by ultrasound. Then 25 μl of the ink was spread onto the surface of glassy carbon by a micropipette and dried under room temperature. The final loading for the catalysts and 40% Pt/C electrocatalysts on work electrode is 0.5 mg cm$^{-2}$. HER tests were conducted in an Ar-saturated 0.5 M $H_2SO_4$ electrolyte at 25 °C. The potential range was from 0 to $-1.0$ V (versus Ag/AgCl) and the scan rate was 2 mV s$^{-1}$. Before measurements, the samples were repeatedly swept from $-0.4$ to 0.3 V (versus Ag/AgCl) in the electrolyte until a steady voltammogram curve was obtained. All the final potentials have been calibrated with respect to a reversible hydrogen electrode (RHE).

**DFT calculations.** All theoretical calculations were performed using Vienna *ab initio* simulation packages (VASP)[40] with projector-augmented wave scheme[41]. The generalized gradient approximation with the Perdew–Burke–Ernzerhof (PBE)[42] functional was used for the exchange–correlation interaction. The plane wave cutoff was set to 400 eV. The convergence of total energy and forces were set to $1 \times 10^{-5}$ eV and 0.05 eV Å$^{-1}$, respectively. A periodically repeated single-layer $MoS_2$ (a trilayer unit of S-Mo-S as a single layer[43]) crystal model with a 20 Å vacuum space has been built for DFT calculations. The Brillouin zone was sampled by a $3 \times 3 \times 1$ k-point grid with the Monkhorst–Pack scheme[44] for structural optimization and a $6 \times 6 \times 1$ k-point grid for electronic structure calculations. More details see the Supplementary Methods.

**Data availability.** The data that support the findings of this study are available from the corresponding authors on request.

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

## Acknowledgements

We gratefully acknowledge the financial support from the Ministry of Science and Technology of China (No. 2016YFA0204100 and 2016YFA0200200), the National Natural Science Foundation of China (No. 21573220 and 21621063), the Key Research Program of Frontier Sciences of the Chinese Academy of Sciences (No. QYZDB-SSW-JSC020), the strategic Priority Research Program of the Chinese Academy of Sciences (No. XDA09030100). We thank staff at the BL14W1 beamline of the Shanghai Synchrotron Radiation Facilities (SSRF) for assistance with the EXAFS and XANES measurements. We also acknowledge the computational resources from National Supercomputing Center in Shenzhen.

## Author contributions

X.B. and D.D. conceived the project and designed the experiments. J.D. performed the materials synthesis, materials characterization and electrochemical measurements. H.L. conducted the DFT calculations. S.W. assisted with the materials characterization. D.D. and M.C. performed the XPS measurements. C.L. and Z.T. conducted the Raman measurements. C.M. performed the 3D tomography. Z.T. and K.S.N. gave the valuable discussions and suggestions. J.D., D.D. and X.B. co-wrote the manuscript.

## Additional information

**Competing interests:** The authors declare no competing financial interests.

