## [Peer Review File · Nature Communications]

Reviewers' Comments:

Reviewer #1 (Remarks to the Author)

The paper presents results for a new nano-structuring strategy to increase the number of edge sites in MoS₂. It also presents results suggesting additional effect of doping with Co.

I find the paper interesting and it could become publishable in Nature Comm. I would suggest that the authors consider the following comments:

1. Tone down the heavily inflated language.
2. Shorten the paper considerably; the basic idea and results are quite simple and will come through more clearly by a concise presentation
3. The overpotential is comparable to the best reported in the literature, not "outperforming" them, since that would indicate something much better. Please substantiate the suggestion that the Co is incorporated in the basal planes and not at the edges.

Reviewer #2 (Remarks to the Author)

The authors reported the multiscale structural and electronic control of MoS₂ foam for highly efficient hydrogen production. The developed 3-dimensional mesoporous MoS₂ foam with chemical doping of Co atoms showed high HER activity and good stability, which demonstrated by Density functional theory calculations. The materials were characterized in detailed and the corresponding catalytic properties were systematically studied. This is a very interesting work, and could be extended to the other new materials and energy-related process. The manuscript could be published after minor revision.

Comments :

1. Is there change for the structure after doping with Co atoms?
2. Some relative literatures such as Adv. Funct. Mater. 2013, 23, 5326, etc. need be added.

Reviewer #3 (Remarks to the Author)

The manuscript contributed by Deng et al. reports an very interesting MoS₂ containing multiscale control in terms of morphological structure and electronic structure. Such 3D mesoporous MoS₂ foam was synthesized with the use assembled SiO₂ photonic crystals as templates. The resulting MoS₂ foam exhibit unique structures at three different length scales: 1) the macroscopic mesoscale pores with size of 30 nm provide channels for the transport of H₃O⁺ and H₂; 2) the nanoscale orientation of MoS₂ nanosheets increases the number of edges to provide more catalytically active sites; 3) the atomic level control over doping changes electronic structures to benefit HER activity. Synergistic effects on these multiscale control make the reported MoS₂ foam to exhibit HER catalytic activity superior to most MoS₂.

TEM image shown in Figure 2g highlights the presence of many edge features. What parameter causes this well-aligned structure? What structure should be observed if the sample is rotated by 90 degrees? It is suggested to take a 3D tomography to highlight the anisotropic orientation of the MoS₂ nanosheets.

HF was used to remove the SiO₂ templates. Did HF etch influence the structure of MoS₂?

Response to the reviewers' comments

Reviewer # 1

The paper presents results for a new nano-structuring strategy to increase the number of edge sites in MoS₂. It also presents results suggesting additional effect of doping with Co.

I find the paper interesting and it could become publishable in Nature Comm. I would suggest that the authors consider the following comments:

Comment: 1) *Tone down the heavily inflated language.*

Response: Thanks for the reviewer's comment. We have modified the language to avoid some inflated expression in the revised manuscript.

Comment: 2) *Shorten the paper considerably; the basic idea and results are quite simple and will come through more clearly by a concise presentation.*

Response: According to the reviewer's suggestion, we have shortened the revised manuscript reasonably and integrally to be more concise.

Comment: 3) *The overpotential is comparable to the best reported in the literature, not "outperforming" them, since that would indicate something much better. Please substantiate the suggestion that the Co is incorporated in the basal planes and not at the edges.*

Response: According to the reviewer's suggestion, we have replaced the description "outperforming" by "comparable" in the revised manuscript (line 10-11, paragraph 1, page 1; line 7-8, paragraph 1, page 3; line 11-12, paragraph 1, page 10; line 14-15, paragraph 2, page 13).

Our recent work have demonstrated that a series of transition metal atoms can be successfully doped into the basal planes of MoS₂, including Pt, Co and Ni etc. (Energy Environ. Sci. 2015, 8, 1594). The prepared method of Co doped mPF-MoS₂ in this manuscript is similar to the previous one by one-pot homogeneous chemical reaction, except that the SiO₂ nanospheres were used as the mesoporous template. Therefore, owing to the similar synthesized process and 2D basal planes are the dominant domains, most Co atoms should be doped into the basal planes instead of the edges. Nevertheless, unlike the Pt dopants in

previous study, Co atoms within the MoS₂ matrix can hardly be distinguished from Mo atoms by spherical aberration-corrected HAADF-STEM to observe their positions. Furthermore, additional DFT calculations also showed Co dopant within the basal plane led to lower surface energy (γ) of 0.35 eV compared to at the edge (S-edge is 0.56 eV and Mo-edge is 0.37 eV), suggesting that doping Co atom into the basal plane is more preferred. Considering few Co atoms at the edges may make contribution to the activity enhancement, we modified the expression about the effect of Co dopants to be more reasonable in the revised manuscript (line 3-5, paragraph 1, page 3; line 7-8, paragraph 1, page 10).

Reviewer # 2

The authors reported the multiscale structural and electronic control of MoS₂ foam for highly efficient hydrogen production. The developed 3-dimensional mesoporous MoS₂ foam with chemical doping of Co atoms showed high HER activity and good stability, which demonstrated by Density functional theory calculations. The materials were characterized in detailed and the corresponding catalytic properties were systematically studied. This is a very interesting work, and could be extended to the other new materials and energy-related process. The manuscript could be published after minor revision.

Comment: 1) *Is there change for the structure after doping with Co atoms?*

Response: Thanks for the reviewer's comment. When the Co doping contents were below 16.7%, the mesoporous MoS₂ foam has been well retained according to the SEM (Figure S9), TEM (Figure S10a) and HAADF-STEM (Figure 4a) images, and MoS₂ also assembles as vertically aligned layers around the mesopores according to the HRTEM image (Figure S10b). In addition, the Co dopants bring indiscernible chemical state variation of the MoS₂ framework according to the XPS spectra (Figure S11) and Mo K-edge XANES spectra (Figure S12). However, when the Co doping contents exceeded 16.7%, structural variation has happened in the mesoporous MoS₂ foam, according to the Raman spectra showing the E_{2g}¹ and A_{1g} modes of MoS₂ changed significantly (Figure 4e) and the XRD patterns showing other new phases appeared (Figure 4f). Meanwhile, distinct change in pore structure of MoS₂ foam appeared when the Co doping contents exceeded 16.7% (Figure S13). Therefore, there is an optimum doping content (16.7% from our experience) which will provide significant Co

contents but still preserving the integrated mesoporous vertically aligned framework. The change for the structure of MoS₂ after doping with Co atoms has been highlighted in yellow in the revised manuscript (line 6-13, paragraph 1, page 9).

Comment: 2) Some relative literatures such as *Adv. Funct. Mater.* 2013, 23, 5326, etc. need be added.

Response: Thank the reviewer for providing us the literature, and it has been cited in the revised manuscript as Reference 21.

Reviewer # 3

The manuscript contributed by Deng et al. reports an very interesting MoS₂ containing multiscale control in terms of morphological structure and electronic structure. Such 3D mesoporous MoS₂ foam was synthesized with the use assembled SiO₂ photonic crystals as templates. The resulting MoS₂ foam exhibit unique structures at three different length scales: 1) the macroscopic mesoscale pores with size of 30 nm provide channels for the transport of H₃O⁺ and H₂; 2) the nanoscale orientation of MoS₂ nanosheets increases the number of edges to provide more catalytically active sites; 3) the atomic level control over doping changes electronic structures to benefit HER activity. Synergistic effects on these multiscale control make the reported MoS₂ foam to exhibit HER catalytic activity superior to most MoS₂.

Comment: 1) TEM image shown in Figure 2g highlights the presence of many edge features. What parameter causes this well-aligned structure? What structure should be observed if the sample is rotated by 90 degrees? It is suggested to take a 3D tomography to highlight the anisotropic orientation of the MoS₂ nanosheets.

Response: Thanks for the reviewer's comment and suggestion. The well-defined colloidal SiO₂ nanospheres template is the important parameter for the formation of well-aligned MoS₂ layers. Above the SiO₂ surface, the diffusion along the aligned layers is expected to be much faster than the diffusion across the layers, resulting in the oriented MoS₂ nanosheets with many edge features. Without the restriction of SiO₂ substrate, the diffusion process will be isotropic, leading to random-oriented MoS₂ nanosheets (Figure S3).

According to the reviewer's comment, we have performed 3D tomography in HAADF-STEM mode. The bright-field TEM and HAADF-STEM images (Figure S4 and S5) taken at different tilting angles show the projected structures of the mesoporous MoS₂ foam along different orientations, indicating that the mesopores are interconnected throughout the entire 3D MoS₂ framework. Furthermore, the video reconstructed from the tilting series images shows more clearly the 3D structure of the mesoporous MoS₂ foam. We have added the TEM and HAADF-STEM images into Supporting Information of the revised manuscript as Figure S4 and S5, and one sentence about the description of 3D MoS₂ foam in the revised manuscript (line 5-7, paragraph 1, page 5). In addition, the video has been uploaded as a single supporting material.

Comment: 2) *HF was used to remove the SiO₂ templates. Did HF etch influence the structure of MoS₂?*

Response: The HF has already been used to remove silica template when preparing mesoporous MoS₂ in previous works (J. Am. Chem. Soc. 2007, 129, 9522; Adv. Energy Mater. 2012, 2, 970), and the intrinsic structure of MoS₂ has not been influenced by the etching process because the MoS₂ can not be dissolved by HF solution. Therefore, HF was chosen here to remove SiO₂ nanospheres. In order to observe the structure of MoS₂ before and after the removal of SiO₂ template, we carried out the additional HRTEM and XRD experiments. As shown in the HRTEM image (Figure S1a), the MoS₂ layers were grown around the SiO₂ nanosphere and almost all the MoS₂ layers presented as oriented alignment. After etching the SiO₂ nanosphere, the mesopore has been well remained deriving from the residual spaces and vertically aligned MoS₂ layers with a large fraction of exposed edge sites were obtained (Figure 2g). Furthermore, no obvious difference among the diffraction peaks of MoS₂ in the XRD patterns were found before and after the removal of SiO₂ template (Figure S1b). In addition, Raman spectra (Figure S6), XPS spectra (Figure S7) and XANES spectra (inset of Figure 2i) showed no obvious difference of the MoS₂ prepared with or without SiO₂ and HF etching. So combined literatures and our experimental results, it is believed that the MoS₂ is stable within the HF solution. We have added a sentence in the revised manuscript to clarify the influence of HF etching on the structure of MoS₂ (line 6-7, paragraph 1, page 4).

Reviewers' Comments:

Reviewer #2 (Remarks to the Author)

The manuscript has been revised and could be published on Nature Communications.

Reviewer #3 (Remarks to the Author)

The comments and questions are clearly addressed in the revised manuscript. It is ready to be publish.